# Comprehensive Analysis of the Impact of Weight Loss Thresholds on Mouse Models of Fatal Viral Infection

**DOI:** 10.3390/v17091225

**Published:** 2025-09-07

**Authors:** Devin Kenney, Mao Matsuo, Giulia Unali, Alan Wacquiez, Mohsan Saeed, Florian Douam

**Affiliations:** 1Department of Virology, Immunology and Microbiology, Boston University Chobanian and Avedisian School of Medicine, Boston, MA 02118, USA; kenneydj@bu.edu (D.K.); mmatsuo@bu.edu (M.M.); gunali@bu.edu (G.U.); 2National Emerging Infectious Diseases Laboratories, Boston University, Boston, MA 02118, USA; wacquiez@bu.edu (A.W.); msaeed1@bu.edu (M.S.); 3Department of Biochemistry and Cell Biology, Boston University Chobanian and Avedisian School of Medicine, Boston, MA 02118, USA

**Keywords:** mouse models of viral infection, weight loss, fatal viral infection, SARS-CoV-2, C57BL/6, K18-hACE2, BALB/c, euthanasia criteria, influenza A virus, Powassan virus

## Abstract

Preclinical studies in virological research are pivotal to comprehend mechanisms of viral virulence and pathogenesis and evaluate antiviral therapies or vaccines. Mouse models, through access to various genetic strains and amenable reagents, along with their ease of implementation and cost-effectiveness, remain the gold standard for establishing go/no-go thresholds before advancing to non-human primate or clinical studies. In preclinical mouse studies, standardized weight loss thresholds (WLTs)—which correspond to an established percentage of weight change at which animals are humanely euthanized—are a routine metric to quantitatively evaluate the lethality of a viral pathogen and the effectiveness of antiviral countermeasures in preventing fatal viral disease. While it is recognized that WLTs can significantly impact the assessment of viral virulence, they are often established to meet existing ethical or methodological requirements, rather than being based on a specific scientific rationale. Here, we examine how various experimental variables—including mouse and viral strains and the sex ratio within a mouse cohort—influence the ability of a WLT to support the generation of robust mouse models of fatal viral infection. Using various mouse strains and viral pathogens, we report that variations in experimental conditions in mouse preclinical studies can significantly compromise the performance of a non-adjusted WLT to yield an accurate estimate of viral virulence. Our findings advocate for a robust adjustment of WLT to each experimental framework and associated variables to establish mouse models of fatal viral infection that can generate high-resolution data acquisition while upholding ethical standards. Overall, our study provides methodological insights to enhance the unbiased acquisition and benchmarking of viral virulence and antiviral efficacy data in mouse models.

## 1. Introduction

Small animal models, such as mice and hamsters, are critical experimental platforms to evaluate viral virulence, develop antiviral therapeutics, and test vaccine efficacy. A commonly utilized metric for assessing *in vivo* viral virulence and the effectiveness of antiviral countermeasures is the survival rate of animals upon inoculation with a viral pathogen, and how a specific treatment or vaccine impacts survival. While infection can result in animals being found dead in cages (fdic), the survival rate is also heavily dependent on specific clinical criteria that mandate euthanasia. These criteria are crucial for establishing ethical and humane endpoints for animals suffering from severe disease. The most standard criterion is the weight loss threshold (WLT), which corresponds to an institutionally defined percentage of weight change (through the submission and approval of an animal study protocol by a research institution) at which an animal needs to be humanely euthanized.

In mouse studies, WLTs historically range between 20% and 30% weight loss. The use of a specific WLT in a laboratory mainly stems from pre-existing ethical standards in the institution that hosts the laboratory and/or the historical methodological standards of that laboratory research program. More rarely are WLTs set up based on a specific scientific rationale. However, some studies have advocated for the importance of setting up WLTs depending on the experimental model and framework being used. For instance, in sepsis models of infection, reducing the WLT from 30% to 20% would have resulted in the euthanasia of animals that would have otherwise survived [1], confounding survival rates. This highlights the importance of defining the WLT in accordance with the experimental model and framework being used. Some virus-specific fields, such as in influenza A virus (IAV) research, have also historically recognized that using a high WLT (i.e., 30%) allows for a more accurate characterization of microbial virulence [2,3,4,5]. Nevertheless, many other virus-specific research fields, such as those investigating mosquito-borne orthoflaviviruses, have historically applied a standard 20% WLT without inquiring about its impact on survival rates and disease dynamics assessment [6,7,8,9,10,11,12,13].

The lack of WLT optimization can be problematic in a pandemic-related public health emergency. Applying a non-optimally adjusted WLT can result in an inaccurate characterization of the virulence of a given variant and/or the underestimation of the potency of a drug or vaccine that would protect from death (but not disease), delaying rapid scientific progress and life-saving discoveries. Divergent WLTs across laboratories can also create lab-to-lab reproducibility issues. The SARS-CoV-2 pandemic exposed some of these issues. A non-exhaustive, randomized literature search of SARS-CoV-2 mouse studies (encompassing 26 studies in total [14,15,16,17,18,19,20,21,22,23,24,25,26,27,28,29,30,31,32,33,34,35,36,37,38,39]) revealed that only ~42% of the studies examined (11/26 studies) explicitly reported the WLT used. Among these, 27% employed a WLT of 20%, while 11.5% and 4% used a WLT of 25% and 30%, respectively. The disproportionate use of a 20% WLT over 25% and 30% may undermine major phenotypical features of a viral variant, drug, or vaccine. This reinforces the importance of advocating for new methodological standards aimed at adapting the WLT to specific experimental models and frameworks to generate reproducible and accurate preclinical data, which could represent life-saving opportunities during a viral pandemic. The rapid reporting of mouse studies throughout the pandemic highlighted the importance of adequately stating the WLT to ensure consensus and reproducibility between laboratories using similar model systems. Notably, out of the 26 randomized SARS-CoV-2 mouse studies examined in our literature search, 56% (15/26) did not explicitly mention the WLT used as a euthanasia criterion in their respective methods sections.

Although some studies have reported the importance of using an adequate WLT to perform a comprehensive assessment of microbial virulence in mice [1], the influence of the experimental context and associated variables—including mouse strains, viral variants of a specific pathogen, and the sex ratio—on the ability of WLT to yield robust mouse models of fatal microbial infection has not been comprehensively evaluated. Using SARS-CoV-2 and influenza A (IAV) viruses, we report that WLT should be appropriately increased to 30% to reduce bias introduced by several experimental variables, including mouse strains, genetic variants of the same viral pathogen, and the sex ratio within a mouse cohort. We further demonstrate that 20% WLT is inadequate for capturing sex-specific differences in viral virulence. Our findings emphasize the importance of optimizing the WLT within each experimental framework to create robust mouse models of fatal viral infection capable of accurately assessing viral virulence. Our work also stresses the crucial importance of transparently communicating the WLT and other animal death-related information (e.g., fdic vs. humanely euthanized) to facilitate better inter-lab interpretation and reproducibility. Together, we hope that this work will foster improved methodological standards for evaluating viral virulence and preclinical testing of antiviral countermeasures.

## 2. Methods

### 2.1. Institutional Approvals

All experiments in this study, including those conducted in BSL-3, were approved by Boston University Institutional Biosafety Committee. Animal experiments described in this study were performed in accordance with protocols that were reviewed and approved by the Institutional Animal Care and Use and Committee at Boston University. All mice were maintained in facilities accredited by the Association for the Assessment and Accreditation of Laboratory Animal Care (AAALAC) and approved under protocol PROTO202000020. All replication-competent SARS-CoV-2 experiments were performed in a biosafety level 3 laboratory (BSL-3) at Boston University’s National Emerging Infectious Diseases Laboratories (NEIDL).

### 2.2. Mammalian Cell Lines

African green monkey kidney Vero E6 cells (ATCC^®^ CRL-1586™, American Type Culture Collection, Manassas, VA, USA), human hepatoma Huh7.5 cells (kind gift of Charles M. Rice), and Caco-2 cells co-expressing human angiotensin-converting enzyme 2 (hACE2) and Transmembrane protease, serine 2 (hTMPRSS2) were maintained in Dulbecco’s minimum essential medium (DMEM; Gibco, Carlsbad, CA, USA [#11995-065]) containing 5% fetal bovine serum (FBS, R&D Systems, Minneapolis, MI, USA) and 1% penicillin and streptomycin (100 U/mL and 100 μg/mL) (Gibco, Carlsbad, CA, USA [#5140122]. Caco-2 hACE2/hTMPRSS2 cells were additionally maintained with 2.5 ug/mL each of puromycin and blasticidin for selection of the transgenes. Human Embryonic Kidney (HEK) 293T (ATCC^®^ CRL-11268™) cells and Madin–Darby Canine Kidney (MDCK) (ATCC^®^ CCL-34™) cells were maintained in DMEM and MEM mediums (Gibco), respectively, in the presence of 5% FBS and 1% penicillin and streptomycin. All cell lines were incubated at 37 °C and 5% CO2 in a humidified cell culture incubator.

### 2.3. SARS-CoV-2 Viral Stock Preparation and Titration

All replication-competent SARS-CoV-2 experiments were performed in a biosafety level 3 laboratory (BSL-3) at Boston University’s National Emerging Infectious Diseases Laboratories (NEIDL). SARS-CoV-2 MA30 was generously provided by the laboratory of Dr. Stanley Perlman (University of Iowa). SARS-CoV-2 Delta and Omicron BA.1 were generously provided by the laboratories of Dr. John Connor (Boston University). To generate passage 3 (P3) of the MA30 virus, Vero E6 cells were infected with a multiplicity of infection (MOI) of 0.1 of a P2 stock. To generate the P2 stock of the clinical isolate virus (Delta and BA.1) of SARS-CoV-2, a P1 stock was expanded on Vero E6 (Delta virus) or Caco-2 hACE2/hTMPRSS2 (BA.1) cells by infecting at an MOI of 0.1. For infections, viral adsorption was allowed to occur for 1 h in 10 mL of OptiMEM + GlutaMAX media (Gibco, Carlsbad, CA, USA [51985034]) prior to the addition of 15 mL of DMEM with 5% FBS and 1% Penn/Strep. After 12–16 h (overnight), the media was removed and fresh DMEM 2% FBS was added to the cells. Cell culture media containing viruses were then harvested when cytopathic effect (CPE) was observed. Viral stocks were then concentrated and purified via ultracentrifugation over a 20% sucrose cushion (Sigma-Aldrich, St. Louis, MO, USA). Pellets were then resuspended overnight at 4 °C in 1× PBS and titrated by plaque assay on Vero E6 (MA30 and Delta) or Caco-2 hACE2/hTMPRSS2 cells (BA.1).

For plaque assay, serial dilutions of concentrated and non-concentrated stocks were generated and then 300 μL of each dilution was plated onto 12-well plates and incubated at 37 °C with 5% CO_2_ for 1 h, followed by the addition of 1 mL of 1.2% Avicel in DMEM containing 2% FBS and 1% Penn/Strep. Cells were incubated for 3 days before the overlay was removed; cells were then fixed with 10% neutral-buffered formalin and stained with crystal violet (0.1% crystal violet in 10% ethanol/water).

### 2.4. Influenza A/Puerto Rico/8/1934 (H1N1) Viral Stock Preparation and Titration

A/Puerto Rico/8/1934 (H1N1) (PR8) virus was generated in a BSL-2 lab using reverse genetics as previously described. Briefly, 500 ng of each of the 8 bi-directional plasmids (coding, respectively, for PB2, PB1, PA, HA, NP, NA, M, and NS; a kind gift from Mustapha Si-Tahar, INSERM, University of Tours, France) [40] were transfected using Fugene HD (Promega, Madison, WI, USA) into a co-culture of HEK-293T (4 × 10^5^ cells) and MDCK (3 × 10^5^ cells) cells seeded into a 6-well plate. After 24 h, the media was replaced by 3 mL of fresh media (without FBS) containing 1 µg/mL of trypsin-TPCK (L-(tosylamido-2-phenyl) ethyl chloromethyl ketone) (Worthington Biochemicals, Lakewood, NJ, USA). The cytopathic effect (CPE) was monitored every day to track viral replication. Supernatants were collected after day 3 post-transfection, then briefly centrifuged (5 min × 5000× *g* at 4 °C) and passed through a 0.45 µm filter before being stored at −80 °C until viral titration. MDCK cells were used for plaque-forming assays to measure the PR8 viral titer. Briefly, 30 µL of PR8 stock was serially diluted in 270 µL of Opti-MEM, then each dilution was plated onto 12-well plates and incubated at 37 °C with 5% CO_2_ for 1 h, followed by the addition of 1 mL of 1.2% Avicel mixed with 1× MEM (without FBS) containing 1 µg/mL of trypsin-TPCK [41]. Cells were incubated for three days before the overlay was removed. Cells were then fixed with 10% neutral buffered formalin and stained with crystal violet (0.1% crystal violet in 10% ethanol/water).

### 2.5. Powassan Virus (LB Strain) Viral Stock Preparation and Titration

The Powassan virus LB strain was acquired from the World Reference Center for Emerging Viruses (WRCEVA) at the University of Texas Medical Branch (UTMB). All replication-competent Powassan virus experiments were performed in a biosafety level 3 laboratory (BSL-3) at Boston University’s National Emerging Infectious Diseases Laboratories (NEIDL). To generate stocks of POWV-LB (001v-EVA124), VeroE6 cells were infected with a multiplicity of infection (MOI) of 0.1 and monitored for cytopathic effects (CPE) over the course of 7 days. For infections, viral adsorption was allowed to occur for 2 h in 10 mL of OptiMEM + GlutaMAX media (Gibco, Carlsbad, CA, USA) prior to the addition of 15 mL of DMEM 5% FBS 1% Penn/Strep. After 12-16 h, the media was removed and fresh DMEM 2% FBS was added to the cells. Cell culture media containing viruses were harvested when cytopathic effect (CPE) was observed. Viral stocks were then concentrated and purified via ultracentrifugation over a 20% sucrose cushion (Sigma-Aldrich, St. Louis, MO, USA). Pellets were then resuspended overnight at 4 °C in 1× PBS before being aliquoted and stored at −80 °C.

For plaque assay, Huh7.5 cells were plated in 48-well plates at 2.2 × 10^4^ cells/well, one day prior to infections. Viruses were 10-fold serial diluted in OptiMEM + GlutaMax media, 100 μL of each dilution was added to wells, and incubated for 2 h at 37 °C. After incubation, viruses were removed and 500 μL of a 1:1 mixture of 2× DMEM containing 4% FBS and 20% methylcellulose and incubated for 5 days at 37 °C under 5%CO_2_. After incubation, the overlay media were removed, cells were fixed with 10% neutral-buffered formalin for 2 h, and then stained using 0.1% crystal violet diluted in 10% ethanol.

### 2.6. Mouse Strains

Wild-type C57BL/6J (Jackson Laboratory, catalog # 000664, Bar Harbor, ME, USA), C57BL/6Ntac (Taconic Biosciences, catalog # B6-F and B6-M, Albany, NY, USA), BALB/cJ (Jackson Laboratory, catalog # 000651), and heterozygous K18-hACE2 C57BL/6J (2B6.Cg-Tg(K18-ACE2)2Prlmn/J; Jackson Laboratory, catalog # 034860) were used in these studies. Animals were group-housed by sex in Tecniplast green line individually ventilated cages (Tecniplast, Buguggiate, Italy). Mice were maintained on a 12:12 light cycle at 30–70% humidity and provided ad libitum water and standard chow diets (LabDiet, St. Louis, MO, USA).

### 2.7. SARS-CoV-2 and IAV PR8 Infections in Mice

Male and female 14–22-week-old C57BL/6J, C57BL/6Ntac and BALB/c mice were intranasally inoculated with either 1 × 10^4^ or 1 × 10^5^ plaque-forming units (PFU) of SARS-CoV-2 MA30 in 50 µL of sterile 1× PBS. Male and female 14–22-week-old C57BL/6J and BALB/c mice were intranasally inoculated with either 500 or 1000 PFU of IAV PR8 in 50 μL of sterile 1× PBS. Male and female 13–20-week-old K18-hACE2 mice were intranasally inoculated with either 1 × 10^4^ PFU SARS-CoV-2 Delta (B.1.617.2) or 1 × 10^4^ PFU and 1 × 10^6^ PFU SARS-CoV-2 Omicron (BA.1). All infections were performed while mice were under anesthesia with 1–3% isoflurane.

### 2.8. POWV Infections in Mice

Male and female 18–29-week-old C57BL/6Ntac mice were subcutaneously injected via footpad with 1 × 10^3^ and 5 × 10^3^ PFU of POWV LB in 50 μL of sterile 1× PBS. All infections were performed while mice were under anesthesia with 1–3% isoflurane.

### 2.9. Clinical Scoring and Monitoring for Survival

For survival studies, mice were monitored for changes in body weight, altered respiration, general changes in appearance, responsiveness, and neurological signs of disease. An IACUC-approved clinical scoring system was used to monitor disease progression and define humane endpoints. The score of “1” was given for each of the following situations: body weight, 10–29% loss; respiration, rapid and shallow with increased effort; appearance, ruffled fur and/or hunched posture; responsiveness, low-to-moderate unresponsiveness; and neurological signs, tremors. The sum of these individual scores constituted the final clinical score (0–5). Mice were considered moribund and humanely euthanized in case of weight loss greater than or equal to 30%, or if they received a clinical score of 4 or above for two consecutive days. Body weight and clinical score were recorded once per day for the duration of the study. For the purpose of survival curves, mice euthanized on a given day were counted dead the day after. Mice that were found dead in the cage were counted dead on the same day. For euthanasia, an overdose of ketamine was administered, followed by a secondary method of euthanasia. Additionally, for survival curves, retroactive weight loss thresholds of 20% and 25% were applied to the weight loss criteria for euthanasia and re-graphed to compare the impact of weight loss cut offs.

### 2.10. Statistical Analysis

For all survival curve comparisons, a log-rank (Mantel–Cox) test was performed with *p*-values of <0.05 being statistically significant. For weight loss comparisons, a two-way ANOVA, mixed-effect model with Šídák’s multiple comparison test was used to determine significance. Significance for weight loss is displayed as a “Time × Column” factor to assess significance for weight change trends over time between each group. For peak weight loss comparisons, a one-way ANOVA, Kruskal–Wallis, and uncorrected Dunn’s test (MA30 strain and dose comparison) or unpaired *t*-test with Welch’s correction (Delta and BA.1) was performed. All statistical tests and graphical depictions of results were performed using GraphPad Prism version 10.1.1 software (GraphPad Software, La Jolla, CA, USA). All *p*-values are indicated on graphs.

## 3. Results

### 3.1. Weight Loss Threshold Significantly Impacts the Survival Rates of C57BL/6Ntac Mice upon SARS-CoV-2 Challenge

We initially sought to determine the potential impact of the WLT on the survival rates of mice infected with a commonly used SARS-CoV-2 strain that causes disease in wild-type mice, specifically the mouse-adapted SARS-CoV-2 strain MA30 [36]. We intranasally challenged male and female 13–22-week-old C57BL/6Ntac mice with 1 × 10^4^ or 1 × 10^5^ plaque-forming units (PFU) of SARS-CoV-2 MA30 virus and recorded weight loss and scored additional clinical disease parameters daily over the course of 14 days post-infection (dpi). These parameters included responsiveness, alterations in respiration, behavior, appearance, and neurological symptoms. They were used simultaneously with the WLT to establish humane euthanasia criteria, as further detailed in the Methods Section (see *Clinical Scoring and Monitoring for Survival*). All of these euthanasia criteria were kept consistent throughout all experiments, albeit changing the WLT, to prevent exaggeration of the role of weight loss in driving fatal outcomes. MA30 infection resulted in severe weight loss by 6 dpi, reaching over ~25% in mice challenged with 1 × 10^4^ PFU and ~30% in mice challenged with 1 × 10^5^ PFU (Figure 1A). Both infection doses resulted in clinical scoring in some mice as early as 1 dpi and all mice by 2 dpi, with more severe disease manifesting earlier in mice challenged with 1 × 10^5^ PFU (Appendix A). While no significant differences were observed between a 20%, 25%, and 30% WLT in C57BL/6Ntac survival rates using a 1 × 10^5^ PFU dose, the WLT significantly impacted the survival rates of the animals at a 1 × 10^4^ PFU dose (Figure 1B,C and Appendix A). A 20% WLT resulted in 0% survival, with all mice meeting euthanasia criteria by 7 dpi (Figure 1B). Increasing the WLT to 25% or 30% significantly increased survival to ~29% and ~57%, respectively (Figure 1B). These findings indicate that WLT significantly impacts the survival rates of C57BL/6Ntac mice in a dose-dependent manner.

### 3.2. Genetic Polymorphisms Between Mouse Strains and Sub-Strains Influence How Weight Loss Thresholds Affect Survival Rates During SARS-CoV-2 Infection

Next, we investigated the impact of the WLT on the survival rates of different mouse strains and sub-strains when similarly infected with SARS-CoV-2. We first chose two sub-strains of the C57BL/6 strain, namely C57BL/6Ntac and C57BL/6J, which differ by only 34 single-nucleotide polymorphisms (SNPs) and 2 insertion/deletions (indels) [42,43]. Further relevant to the goal of our study, despite C57BL/6 being the most commonly used inbred mouse strain worldwide, many studies do not necessarily report on the C57BL/6 sub-strain being used. This is despite C57BL/6N mice having an attenuated inflammatory response to dsRNA compared to C57BL/6J [44]. To compare the impact of the WLT between the C57BL/6Ntac and C57BL/6J sub-strains, we infected mice with SARS-CoV-2 MA30 and monitored their body weight and clinical score (disease) over time. When using a 1 × 10^4^ and 1 × 10^5^ PFU viral inoculum, C57BL/6J exhibited a more moderate peak weight loss compared to C57BL/6Ntac mice (Figure 1A,D). This is despite the fact that C57BL/6J reached peak weight loss two days earlier than C57BL/6Ntac (at 4 dpi) and developed disease as early as 1 dpi (Appendix A). Notably, the survival rates of C57BL/6J mice were less affected by the choice of WLT relative to C57BL/6Ntac mice (Figure 1B,E) at a 1 × 10^4^ PFU dose, with 16% survival at 20% WLT, and 33% survival at both 25% and 30% WLT. Regardless of the sub-strains, survivors and non-survivors displayed similar weight loss dynamics during the acute disease phase (Appendix A), suggesting that WLT-dependent differences in survival rates are driven by a differential capacity of the two sub-strains to tolerate severe weight loss and disease. While increasing the infection dose reduced the influence of WLT on survival in C57BL/6Ntac mice (Figure 1B,C), the survival of C57BL/6J mice remained unaffected by the selected WLT at a higher infection dose (Figure 1E,F). To further confirm that WLT does not affect the survival of C57BL/6J mice regardless of dose, we also challenged C57BL/6J mice with lower viral doses (i.e., 5 × 10^3^ and 1 × 10^3^ PFU). Survival and weight loss data from these low-dose infections supported our observation that the WLT has minimal impact on survival rates in C57BL/6J mice relative to C57BL/6Ntac mice (Appendix A).

To further investigate the impact of mouse strains on the ability of the WLT to assess viral virulence, we also included in our comparative panel a more divergent strain, BALB/c, which exhibits a more pronounced CD4 T helper (Th) type II (Th2) immune response than C57BL/6 mice but higher lung inflammatory responses compared to C57BL/6N mice [44,45]. Using a 1 × 10^4^ PFU viral inoculum, BALB/c exhibited 0% lethality regardless of the WLT, unlike C57Bl/6J and Ntac (Figure 1G,H and Appendix A). At a higher viral inoculum (1 × 10^5^ PFU), MA30 infection was similarly fatal in BALB/c mice and C57BL/6J mice when using a 20% (16% survival) or 25% (33% survival) WLT (Figure 1F,I and Appendix A), with C57BL/6Ntac mice still showing the highest susceptibility to infection at this dose (0% survival) (Figure 1C). The impact of increasing WLT from 20% to 25% was moderate in BALB/c mice and similar to that in C57BL/6J mice. Albeit not statistically significant (*p* = 0.0671), a 30% WLT yielded higher BALB/c survival (60% survival) compared to a 20% WLT, an effect observed with C57BL/6Ntac but not C57BL/6J mice (Figure 1C,F,I). At that viral dose and with a 30% WLT, the survival rate of BABL/c mice was also higher than that of the two C57BL/6 sub-strains (Figure 1C,F,I).

Of note, for C57BL/6J and Ntac mouse strains, mice that were found dead in cages (fdic; Figure 1J: diamonds) generally showed lesser weight loss compared to those that experienced severe disease but recovered or had to be euthanized (Figure 1J: circles). Pooling mice from both strains and doses that were fdic or survived, we found that 5/7 (71%) mice fdic had not reached a 20% WLT, while mice reaching a 25% weight loss were never fdic (0/11). These findings further underscore how weight loss as a clinical sign of disease is not necessarily coupled with lethality. Mouse sex ratio within each experimental condition also did not appear to influence the impact of the WLT on survival rates (Figure 1J).

Collectively, our data highlight how genetic variations between mouse strains and sub-strains can influence the impact of the WLT on survival rates and, consequently, the establishment of mouse models of fatal viral infection capable of providing a robust assessment of viral virulence.

### 3.3. Viral Genetic Variants Influence the Impact of Weight Loss Thresholds on Survival Rates During SARS-CoV-2 Infection

We then investigated how SARS-CoV-2 clinical variants associated with differential virulence in humans affect the impact of the WLT on mouse survival rates. To do so, we utilized a C57BL6/J transgenic mouse model expressing human angiotensin-converting enzyme 2 (hACE2), the cell entry receptor of SARS-CoV-2, a widely used mouse model in the *Sarbecovirus* field because of its susceptibility to clinical isolates [23,27,28,46]. Significantly, while lethality in MA30-infected mice is linked to rapid, severe pulmonary disease [36], lethality in K18-hACE2 mice upon infection with SARS-CoV-2 clinical isolates is associated with severe neuroinvasion [18,47]. Consistent with this pattern, the SARS-CoV-2 Delta (B.1.617.2) variant causes severe disease and viral neuroinvasion associated with lethality in K18-hACE2 mice at a standard 10^4^ PFU inoculum dose [47]. However, the Omicron BA.1 variant is attenuated in this model, causing only mild disease and no lethality and neuroinvasion at a similar viral dose [48,49,50]. Here, we intranasally challenged 13–20-week-old male and female K18-hACE2 transgenic mice with Delta (high-virulence) or BA.1 (low-virulence) virus and monitored weight loss, survival, and clinical scoring. Consistent with previous studies, Delta infection resulted in significant weight loss with a 10^4^ PFU dose compared to 10^4^ PFU BA.1 infection, which did not result in any major weight change (Figure 2A). Notably, increasing the inoculum dose of BA.1 to 10^6^ PFU enhanced weight loss, albeit to a significantly lesser extent than Delta 10^4^ PFU infection (Figure 2A).

Although a standard WLT of 20% resulted in a non-lethal infection with BA.1 10^4^ PFU inoculum, increasing the dose of BA.1 to 10^6^ PFU led to levels of lethality comparable to the highly virulent Delta at a 10^4^ PFU dose (Figure 2B; 33.3% and 12.5% survival rate for BA.1 10^6^ PFU and Delta 10^4^ PFU, respectively). In contrast, a WLT of 25% or 30% significantly increased the survival rate of animals inoculated with 10^6^ PFU of BA.1 (Figure 2B–D; 25% WLT: 58.3% vs. 12.5% and 30% WLT: 75% vs. 25% survival rate, for BA.1 10^6^ PFU and Delta 10^4^ PFU, respectively), “rescuing” the attenuated virulence phenotype of BA.1. Notably, altering the WLT did not affect survival rates upon Delta infection (Figure 2B–D).

These data demonstrate that the WLT has a greater impact on the survival rates associated with SARS-CoV-2 variants that induce mild-to-moderate disease compared to those that result in more severe disease. These findings call for caution when examining the attenuation of viral variants with a standard WLT (20%) applied, as this could underestimate the assessment of their attenuated nature.

### 3.4. Biological Sex Is a Confounding Variable When Establishing Weight Loss Thresholds

The average weight of female mice is well recognized to be significantly lower than that of males, a phenotype also reflected in our data set [51] (Appendix A). In line with this, the body mass required to reach euthanasia criteria is lower in for females, with a greater differential as the percent weight loss increases (Appendix A). This suggests that a low WLT could introduce a misleading sex bias when acquiring survival data, and such an impact has not been well documented. Given that males and females have been reported to exhibit different susceptibility to SARS-CoV-2 infection, with males presenting with more severe disease and a higher incidence of hospitalization and lethal infection [52,53,54,55,56], we evaluated how biological sex impacts the ability of the WLT to generate accurate sex-bias survival rates upon Delta (10^4^ PFU) and BA.1 (10^6^ PFU) infection.

Upon BA.1 infection, we observed a relatively similar weight loss dynamics between males and females, but females recovered more rapidly from weight loss (Figure 3A). The WLT only had a significant impact on the survival rates of female mice, not male mice (Figure 3B,C). Specifically, a female survival rate of 33.3% was recorded with a 20% WLT, while this rate increased to 83.3% with a 30% WLT, suggesting that a low, suboptimal WLT can significantly exacerbate viral virulence in females compared to males.

Unlike BA.1 infection, female mice elicited more severe weight loss than male mice from Delta infection, with females showing an average weight loss below 25% while males remained near 20% (Figure 3D). Notably, altering WLT had no impact on male survival upon Delta infection, but females had a significant increase in survival with a 30% WLT (22.2%) compared to a 20% or 25% WLT (0%) (Figure 3E,F). Supportive of a stronger impact of WLT on female survival, male mice that survived infection with Delta or high-dose BA.1 showed more gradual and less weight loss compared to those that succumbed to infection (Appendix A). However, female mice showed similar weight loss dynamics regardless of infection outcome (death or recovery) (Appendix A).

The mean of individual peak weight loss in females was significantly lower upon Delta challenge than in males (Figure 3G), consistent with a lower average weight baseline (Appendix A). Additionally, no fdic female mice were reported while two males were fdic, collectively emphasizing the greater impact of weight loss on the survival dynamics of female mice. Peak weight loss was similar between male and female BA.1 challenged mice, although both males and females showed highly variable weight loss between individual mice (Figure 3H) accounting for the significant impact of WLT on BA.1-induced survival rates. Of note, there was no major difference between the clinical scoring of male and female mice challenged with Delta (Appendix A) or BA.1 (Appendix A) mice.

Collectively, our findings illuminate how too stringent a WLT likely disfavors the accurate assessment of female survival rates compared to males, leading to misleading sex-biased phenotypes of viral virulence.

### 3.5. Weight Loss Thresholds Significantly Impact Survival Rates upon Influenza A Virus Infection

We next investigated the influence of WLT on mouse survival rates upon infection with another respiratory virus. For this, we used a prototypic influenza A virus (IAV) (Puerto Rico/8/1934; PR8), which infects wild-type, inbred mouse strains. We intranasally inoculated C57BL/6J mice with 500 and 1000 PFU and monitored disease using a similar protocol to that used for SARS-CoV-2. At a lower infectious dose (500 PFU), we observed moderate weight loss and only one out of ten mice met euthanasia criteria when using a 20% WLT. No fatalities occurred with a 25% or 30% WLT (Figure 4A,B and Appendix A). When using a higher viral dose (1000 PFU), this trend was exacerbated. While three out of ten (30%) mice met euthanasia criteria with a 20% WLT (Figure 4A,C and Appendix A), no fatalities were observed with a 25% or 30% WLT, though these differences did not reach significance (Figure 4B,C). These findings are consistent with our SARS-CoV-2 data, suggesting that increasing the WLT to 30% can provide a more resolutive assessment of viral virulence in the context of respiratory infection.

To assess whether these findings were not specific to C57BL/6J, we also inoculated BALB/c mice with 1000 PFU of IAV PR8. Mean weight loss peaked at 8 dpi and was above the 20% WLT, similar to C57BL/6J mice (Figure 4A,D). However, using a 20% WLT, we observed 44% survival in BALB/c mice, which was significantly reduced to 0% survival with a 25% or 30% WLT (Figure 4E and Appendix A). The mean of individual peak weight loss for each mouse strain and viral dose mirrored these findings (Figure 4F). At a 1000 PFU dose, mouse sex ratio also did not appear to influence the impact of the WLT on survival rates (Figure 4F). Collectively, these results further support the impact of the WLT on survival rates during respiratory viral infections in a mouse strain- and viral dose-dependent manner. 

### 3.6. Impacts of Weight Loss Thresholds on Non-Respiratory Viral Infections

To extend our analyses to non-respiratory viral infections, we performed infections with Powassan virus (POWV), a neurotropic tick-borne orthoflavivirus which can cause severe and potentially fatal encephalitis, as well as long-lasting cognitive issues in survivors [57,58]. It has been previously demonstrated that POWV infects inbred, immunocompetent laboratory mice and can cause lethal disease [59]. We subcutaneously inoculated C57BL/6NTac mice via the footpad route with either 1 × 10^3^ or 5 × 10^3^ PFU of POWV. Weight loss dynamics were similar regardless of the viral inoculum, and peak weight loss was similar between the two doses (Figure 5A,B). Notably, despite the weight loss dynamics being similar, a 1 × 10^3^ viral inoculum was more lethal than a 5 × 10^3^ inoculum regardless of the WLT, suggesting that a higher viral dose may trigger a more protective host response. The WLT also had no (1 × 10^3^ PFU) to limited (5 × 10^3^ PFU) impact on survival (Figure 5C,D), despite the WLT having a significant impact on C57BL6/Ntac survival in the context of SARS-CoV-2 infection (Figure 1A–C). Notably, weight loss also did not correlate with lethality (Figure 5B). Similarly to IAV infections, sex distribution among mice in each experimental condition suggested no apparent impact of the WLT on survival rates (Figure 5B).

These findings parallel observations in K18-hACE2 mice infected with SARS-CoV-2 Delta, suggesting that lethality during neuroinvasive/neurotropic infections are not significantly impacted by varying the WLT.

## 4. Discussion

Measuring survival rates after infection in animal models is a gold standard for assessing viral virulence and testing the efficacy of antiviral drugs or vaccines to protect from fatal outcomes. However, these survival rates depend on WLT, which defines the weight loss endpoints upon which animals must be humanely euthanized and recorded as dead. Although the importance of the WLT to accurately measuring viral virulence has been appreciated [1], how various experimental variables influence the ability of WLT to provide an accurate characterization of viral virulence has not been well described. Utilizing SARS-CoV-2 and influenza A virus, our study reports the significant impact of mouse (sub)-strain, viral variant, and biological sex on the ability of the WLT to provide an accurate assessment of viral virulence in mice, and by extension, to support the generation of robust mouse models of fatal viral infection.

In this study, we employed multiple mouse strains and sub-strains to evaluate the influence of mouse genetic background on the ability of the WLT to accurately measure viral virulence. One of our focuses was on two sub-strains of C57BL/6 mice, C57BL/6J and C57BL/6N mice. Historically, wild-type C57BL/6 mice are generally referred to as C57BL/6, BL6, C57, or “black 6” mice in scientific communication. However, there are numerous sub-strains of C57BL/6 mice [43,60], with C57BL/6N and C57BL/6J being the most prominent ones. Both sub-strains originated from the C57BL/6 mice originally bred in the 1920s by C.C. Little, who founded The Jackson Laboratory. C57BL/6J mice now derive from breeding colonies that came from the original C57BL/6 inbred strain at The Jackson Laboratory facilities. The C57BL/6N sub-strain was developed when this C57LB/6 inbred strain was sent to the National Institutes of Health (NIH) in 1951 and subsequently distributed to other facilities [43,61]. The C57BL/6N sub-strain used in this study comes from Taconic Biosciences, which received C57BL/6N mice from the NIH in 1991 [43,61]. Divergent breeding strategies between the C57BL/6J and C57BL/6N sub-strains have resulted over time in behavioral differences and at least 34 distinct single-nucleotide polymorphisms (SNPs) and 2 indels [62,63]. Evidence that increasing the WLT from 20% to 30% significantly alters C57BL/6N survival rates from ~0% to 57% upon 1 × 10^4^ PFU MA30 infection, but not C57BL/6J survival rates, has multiple ramifications. *First*, our findings show that the WLTs established based on the C57BL/6J mouse model are not necessarily compatible with other C57BL/6 sub-strains. *Second*, our results strongly emphasize the importance of precisely reporting the mouse (sub-)strain being used for a given study and its origin to mitigate lab-to-lab conflicting findings. Here, we only tested two sub-strains of C57BL/6 mice, though there are over twenty sub-strains reported [42]. *Third*, there is a growing interest in leveraging mouse genetic diversity and associated SNPs, notably through the Collaborative Cross [25,64,65,66,67] to identify novel host determinants of susceptibility to viral infection and disease; survival curves are the primary readout for such screens. Our data, through the differential viral virulence observed between two minimally divergent C57BL/6J and C57BL/6N mouse sub-strains when using a 30% WLT, illuminate how an optimized WLT can provide novel avenues to identify genetic determinants of viral virulence in vivo.

Beyond mouse (sub-)strains, we also observed an influence of the WLT on the survival rates of a divergent inbred strain, BALB/c, upon MA30 infection, which manifested in a unique fashion relative to C57BL/6 sub-strains. BALB/c are known to mount Th2-biased responses compared to C57BL/6 mice upon infection and to develop stronger lung inflammatory responses compared to C57BL/6N mice [44,45]. We found that BALB/c mice were more resistant to MA30 infection at a 1 × 10^4^ and 1 × 10^5^ PFU viral dose compared to both C57BL/6 sub-strains. While no lethality was observed at 1 × 10^4^ PFU, unlike both C57BL/6 sub-strains, this increased resistance was only conserved when using a 30% WLT upon infection with a 1 ×10^5^ PFU inoculum. At that viral dose, when using a 20% or 25% WLT, the survival rates of BALB/c were similar to those of C57BL/6J mice. Collectively, our BABL/c findings further reinforce evidence of how mouse strains influence the ability of the WLT to predict viral virulence, the potential of an optimal WLT to uncover genetic determinants of viral virulence, and how optimizing the WLT to a given mouse strain is critical to establish resolutive mouse models of fatal viral infection.

Focusing on the pathogen side, our study also demonstrates how WLT more significantly impacts the survival rates associated with SARS-CoV-2 variants, driving mild-to-moderate disease compared to more severe variants. Many viral pathogens, particularly highly transmissible and chronic viruses, display an extensive genetic diversity associated with distinctive virulence and pathogenesis phenotypes [68,69,70,71,72]. Our results suggest that the broad application of a standard WLT to many, if not all, variants of a given pathogen may undermine the accurate appreciation of the virulence of many of these variants and preclude our understanding of the evolution of viral pathogens. Our findings, therefore, stress the importance of evaluating the impact of the WLT for each new viral strain being investigated and transparently reporting such thresholds in scientific communications.

The role of biological sex in regulating susceptibility to viral infection and disease is a growing field of study. Females can mount higher inflammatory responses and have a lower average weight baseline compared to males [51,73,74], all of which could contribute to their more severe weight loss than males upon viral challenge. Our results align with these observations and show that the WLT has a more predominant impact on female survival than on males, suggesting that the WLT should be adjusted to avoid exacerbating any sex-biased differences in viral virulence. These adjustments will become increasingly important as research into sex-specific immune responses in small animal models expands in the coming decades, transforming our understanding of women’s health and driving the development of innovative treatments.

Importantly, our findings also extend beyond SARS-CoV-2, as we also report an influence of the WLT for the assessment of IAV virulence in C57BL/6J and BALB/c strains, with the most significant impact observed in BALB/c. However, the WLT only minimally influenced survival rates upon POWV infection, a neurotropic virus. Notably, these findings are consistent with the lack of WLT influence on the survival rates of K18-hACE2 mice upon Delta infection, highlighting how assessing the virulence of neurotropic viruses is less dependent on the WLT. While it remains unclear whether the influence of the WLT on assessing viral virulence is more substantial for respiratory viral infections compared to other viral pathogens with different tissue targets in mice, these results highlight the importance of tailoring the WLT to each experimental setup to ensure a resolutive measurement of viral virulence.

More broadly, our study has implications for the development of antiviral therapies and vaccines. Prematurely euthanizing animals due to a suboptimal WLT can underestimate the efficacy of antiviral treatments and vaccines against viral virulence and viral disease in general. In emergency response scenarios, an antiviral countermeasure that can prevent lethality could be a critical resource for public health, regardless of its ability to hamper disease entirely. Additionally, an optimized WLT can help to identify promising antiviral countermeasures that could still benefit from major refinements, while a suboptimal WLT could terminate their development prematurely. As our appreciation of post-acute sequelae of viral infection, particularly through long-COVID-19 [75,76,77,78], is growing, optimized WLTs may help to foster the generation of survivors who recovered from acute viral disease, providing novel opportunities to enhance our understanding of post-acute sequelae of viral infections and to treat these diseases.

A significant barrier to adjusting and potentially increasing the WLT to 30% is the institutionally-imposed WLT in animal protocols, which is often established at 20%. However, animal protocol amendments can be submitted to support studies that warrant altering the standard WLT of 20%, that is, upon robust scientific justification. We believe that this study will aid in providing scientific evidence to support such protocol amendments or, at the very least, demonstrate that there is value for institutions in providing exceptional permission for laboratories to test the impact of the WLT on survival rates, thereby justifying the need for a formal protocol amendment. Standardizing a 25 or 30% WLT across institutions could have considerable benefits for reproducibility and/or validation studies, fostering the pace of scientific discoveries.

In conclusion, our study underscores the critical importance of adjusting the WLT based on all variables within a given experimental framework to enable the accurate measurement of viral virulence and antiviral countermeasure efficacies and to foster the generation of robust mouse models of fatal viral infection. It also highlights the need for greater transparency in scientific communications to enhance reproducibility efforts across laboratories. Specifically, we call for researchers to clearly communicate the mouse strains and sub-strains used, their sex ratio, the WLT applied, and the proportion of animals found dead in cages versus those euthanized based on the WLT for each survival assay. To achieve this, we propose the following template, presented in Figure 6. Such practices will strengthen the potential of virological preclinical studies to drive scientific breakthroughs.

## Figures and Tables

**Figure 1 viruses-17-01225-f001:**
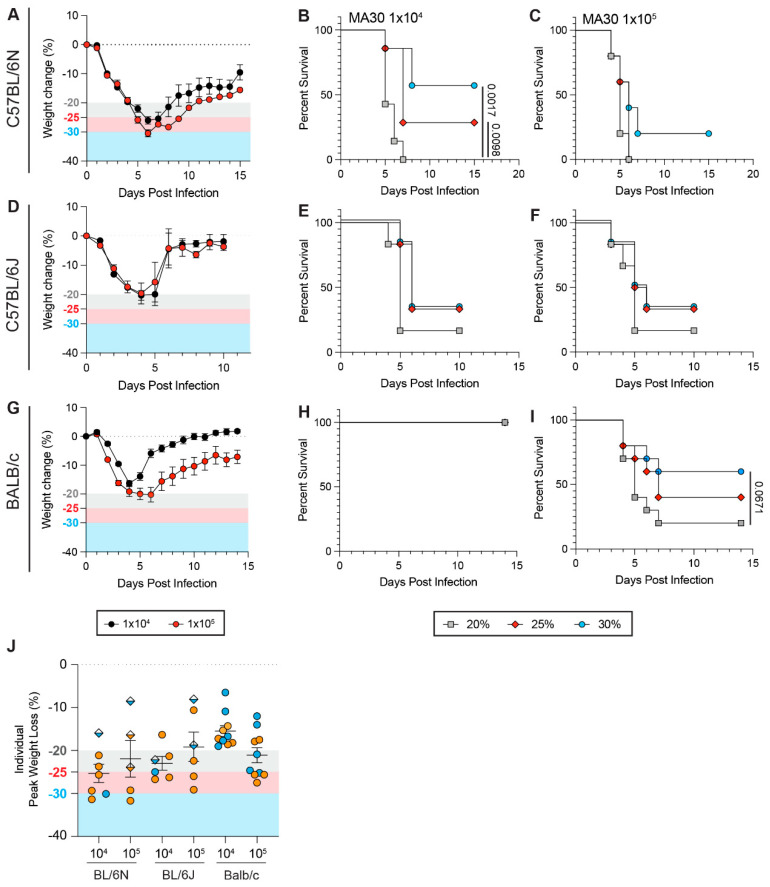
Strain background influences the impact of weight loss thresholds on survival during SARS-CoV-2 infection. In the experiment, 13–22-week-old male and female C57BL/6Ntac (**A**–**C**), C57BL/6J (**D**–**F**) and BALB/c (**G**–**I**) were infected with 1 × 10^4^ (*n* = 7 C57BL/6Ntac, 5 females and 2 males; *n* = 6 C57BL/6J, 4 females and 2 males; *n* = 10 BALB/c, 5 males and 5 females) or 1 × 10^5^ (*n* = 5 C57BL/6Ntac, 4 females and 1 male; *n* = 6 C57BL/6J, 4 females and 2 males; *n* = 10 BALB/c, 5 females and 5 males) plaque-forming unites (PFU) of SARS-CoV-2 MA30 virus. Weight loss and survival were assessed over the course of 15 (C57BL/6Ntac) or 10 (C57BL/6J) days post-infection (dpi). (**A**) Weight loss of C57BL/6Ntac mice infected with 1 × 10^4^ (black) or 1 × 10^5^ (red) PFU of SARS-CoV-2 MA30. (**B**,**C**) Survival curves of C57BL/6Ntac mice challenged with 1 × 10^4^ (**B**) or 1 × 10^5^ (C) with a 20% (gray squares), 25% (red diamonds), or 30% (blue circles) weight loss threshold (WLT) applied. (**D**) Weight loss of C57BL/6Ntac mice infected with 1 × 10^4^ (black) or 1 × 10^5^ (red) PFU of SARS-CoV-2 MA30. (**E**,**F**) Survival curves of C57BL/6Ntac mice challenged with 1 × 10^4^ (**E**) or 1 × 10^5^ (F) with a 20% (gray squares), 25% (red diamonds), or 30% (blue circles) weight loss threshold (WLT) applied. (**G**) Weight loss of BALB/c mice infected with 1 × 10^4^ (black) or 1 × 10^5^ (red) PFU of SARS-CoV-2 MA30. (**H**–**I**) Survival curves of BALB/c mice challenged with 1 × 10^4^ (H) or 1 × 10^5^ (I) with a 20% (gray squares), 25% (red diamonds), or 30% (blue circles) weight loss threshold (WLT) applied. (**J**) Individual peak weight loss of C57BL/6Ntac, C57BL/6J, and BALB/c mice after infection (mean ± SEM). Female and male mice are shown in orange and blue, respectively. Diamonds represent mice that were found dead in cage (fdic). Closed circles represent mice that survived or were euthanized. A Mantel–Cox log-rank test with a 95% confidence interval was applied to determine the significance of survival differences, and for peak weight loss comparisons, a one-way ANOVA, Kruskal–Wallis, and uncorrected Dunn’s test. A two-way ANOVA, mixed-effect model with Šídák’s multiple comparison test was used to determine the significance of weight loss trends over the course of infection. *p*-values are indicated on plots. If no *p*-value is shown between conditions, the difference is not statistically significant.

**Figure 2 viruses-17-01225-f002:**
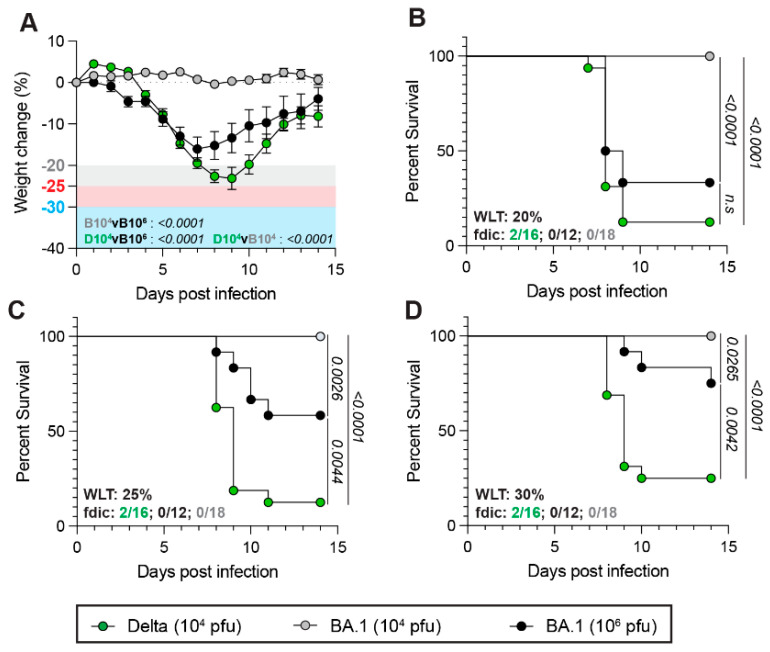
Weight loss thresholds have a more drastic impact on moderate SARS-CoV-2 infection in hACE2 transgenic mice. In the experiment, 12–20-week-old male and female K18-hACE2 transgenic mice were infected with 1 × 10^4^ PFU of SARS-CoV-2 Delta (*n* = 16, 9 females and 7 males) or 1 × 10^4^ (*n* = 18, 9 females and 9 males) and 1 × 10^6^ (*n* = 12, 6 males and 6 females) PFU of Omicron (BA.1). Weight loss and survival were assessed over the course of 14 days. (**A**) Weight loss of K18-hACE2 mice after Delta 1 × 10^4^ (green), 1 × 10^4^ BA.1 (gray), or 1 × 10^6^ BA.1 (black) challenge. (**B**–**D**) Survival curves of K18-hACE2 mice infected with 1 × 10^4^ Delta (green), 1 × 10^4^ BA.1 (gray), or 1 × 10^6^ BA.1 (black) with a 20% (**B**), 25% (**C**), or 30% (**D**) WLT applied. A Mantel–Cox log-rank test with a 95% confidence interval was applied to determine the significance of survival differences. A two-way ANOVA, mixed-effect model with Šídák’s multiple comparison test was used to determine the significance of weight loss trends over the course of infection. *p*-values are indicated on plots. If no *p*-value is shown between conditions, the difference is not statistically significant.

**Figure 3 viruses-17-01225-f003:**
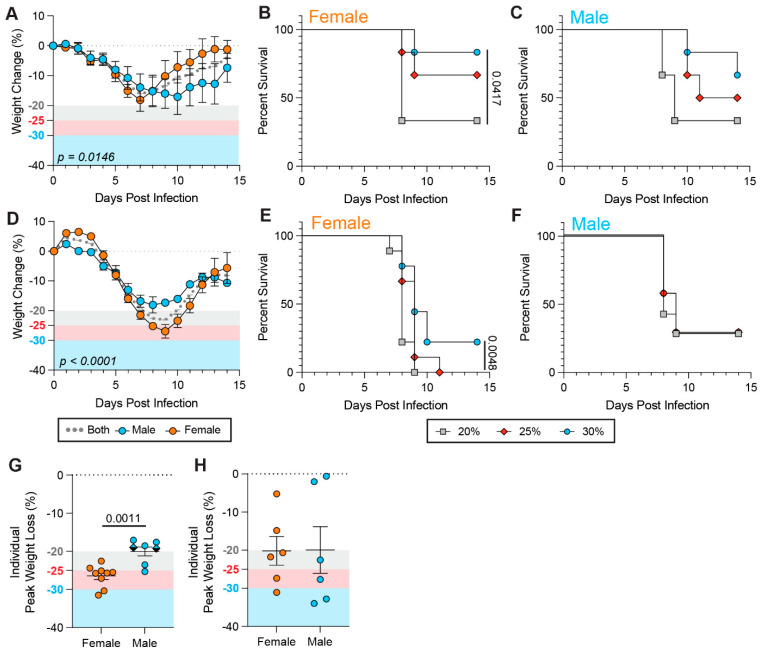
Weight loss thresholds have a greater impact on female mice than on males. (**A**) Weight loss of male (blue), female (orange), and average of male and female (both; gray) during infection with 1 × 10^6^ PFU SARS-CoV-2 Omicron BA.1 (*n* = 6 males and 6 females). Points represent the mean ± SEM. (**B**,**C**) Survival of female (**C**) and male (**D**) K18-hACE2 challenged with 1 × 10^6^ PFU Omicron BA.1. A 20% (gray squares), 25% (red diamonds), or 30% (blue circles) WLT was applied (*n* = 6 males and 6 females). (**D**) Weight loss of male (blue), female (orange), and average of male and female (both; gray) during infection with 1 × 10^4^ PFU SARS-CoV-2 Delta (*n* = 7 males and 9 females). Points represent the mean ± SEM. (**E**,**F**) Survival of female (**E**) and male (**F**) K18-hACE2 challenged with 1 × 10^4^ PFU Delta. A 20% (gray squares), 25% (red diamonds), or 30% (blue circles) WLT was applied (*n* = 7 males and 9 females). (**G**,**H**) Individual peak weight loss (%) in females (orange) and males (blue) during (**G**) Delta and (**H**) Omicron BA.1 infection (mean ± SEM). Diamonds represent mice that were found dead in cage (fdic). Closed circles represent mice that survived or were euthanized. A Mantel–Cox log-rank test with a 95% confidence interval was applied to determine the significance of survival differences, and for peak weight loss comparisons, a one-way ANOVA, Kruskal–Wallis, and uncorrected Dunn’s test. A two-way ANOVA, mixed-effect model with Šídák’s multiple comparison test was used to determine the significance of weight loss trends over the course of infection. *p*-values are indicated on plots. If no *p*-value is shown between conditions, the difference is not statistically significant.

**Figure 4 viruses-17-01225-f004:**
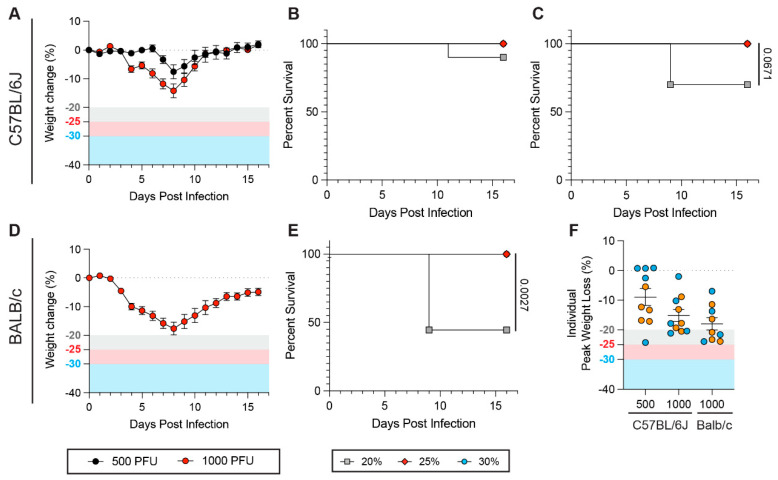
Weight loss thresholds influence mouse survival rates upon influenza A virus infection. In the experiment, 13-22-week-old male and female C57BL/6J (**A**–**C)** and BALB/c (**D**,**E**) were infected with 500 (*n* = 10 C57BL/6J, 5 females and 5 males) or 1000 (*n* = 10 C57BL/6J, 5 females and 5 males; *n* = 10 BALB/c, 5 females and 4 males) plaque-forming units (PFU) of influenza A (IAV) H1N1 PR8 virus. Weight loss and survival were assessed over the course of 16 days post-infection (dpi). (**A**) Weight loss of C57BL/6J mice infected with 500 (black) or 1000 (red) PFU of IAV PR8. (**B**,**C**) Survival curves of C57BL/6J mice challenged with 500 (**B**) or 1000 (**C**) PFU of IAV PR8 with a 20% (gray squares), 25% (red diamonds), or 30% (blue circles) weight loss threshold (WLT) applied. (**D**) Weight loss of BALB/c mice infected with 1000 PFU of IAV PR8. (**E**) Survival curve of BALB/c mice challenged with 1000 PFU of IAV PR8 with a 20% (gray squares), 25% (red diamonds), or 30% (blue circles) weight loss threshold (WLT) applied. (**F**) Individual peak weight loss of C57BL/6J and BALB/c mice after infection (mean ± SEM). Female and male mice are shown in orange and blue, respectively. Closed circles represent mice that survived or were euthanized (no mice were found dead in cages). A Mantel–Cox log-rank test with a 95% confidence interval was applied to determine the significance of survival differences, and for peak weight loss comparisons, a one-way ANOVA, Kruskal–Wallis, and uncorrected Dunn’s test. A two-way ANOVA, mixed-effect model with Šídák’s multiple comparison test was used to determine the significance of weight loss trends over the course of infection. *p*-values are indicated on plots. If no *p*-value is shown between conditions, the difference is not statistically significant.

**Figure 5 viruses-17-01225-f005:**
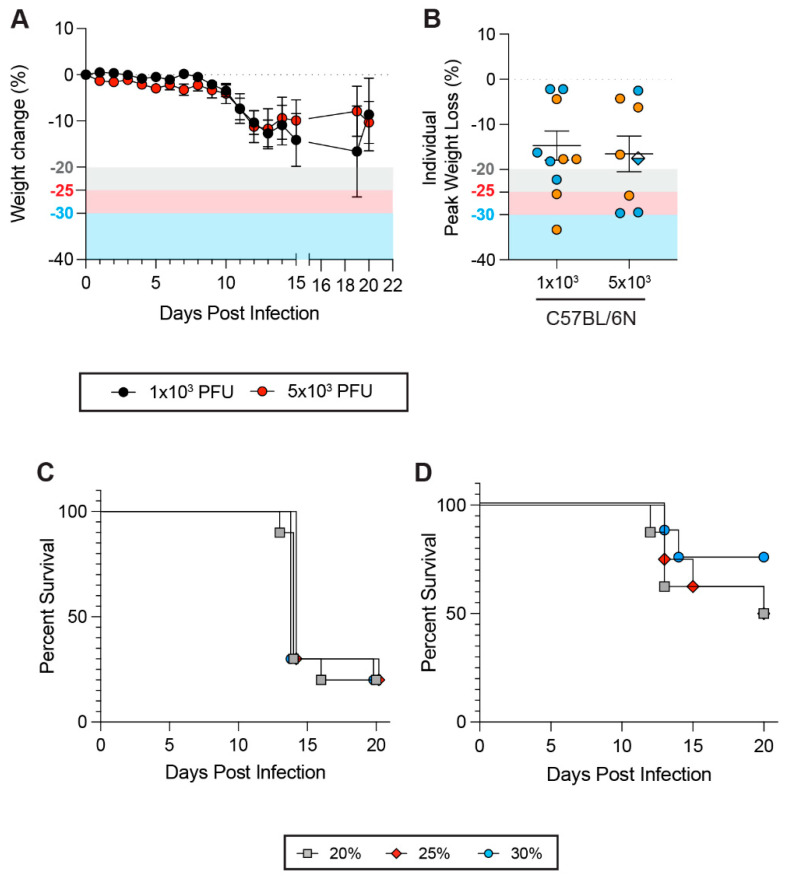
Weight loss thresholds do not impact survival rates upon Powassan virus infection. In the experiment, 18–29-week-old male and female C57BL/6Ntac mice were infected with 1 × 10^3^ (*n* = 10 mice, 5 females and 5 males) or 5 × 10^3^ (*n* = 8 mice, 4 females and 4 males) plaque-forming units (PFU) of POWV LB virus. Weight loss and survival rates were assessed over the course of 16 days post-infection (dpi). (**A**) Weight loss of C57BL/6Ntac mice infected with 1 × 10^3^ (black) or 5 × 10^3^ (red) PFU of POWV LB. (**B**) Individual peak weight loss of C57BL/6Ntac mice after infection (mean ± SEM). Female and male mice are shown in orange and blue, respectively. Diamonds represent mice that were found dead in cage (fdic). Closed circles represent mice that survived or were euthanized. (**C**,**D**) Survival curves of C57BL/6Ntac mice challenged with 1 × 10^3^ (**B**) or 5 × 10^3^ (**C**) PFU of POWV LB with a 20% (gray squares), 25% (red diamonds), or 30% (blue circles) weight loss threshold (WLT) applied. A Mantel–Cox log-rank test with a 95% confidence interval was applied to determine the significance of survival differences, and for peak weight loss comparisons, a one-way ANOVA, Kruskal–Wallis, and uncorrected Dunn’s test. A two-way ANOVA, mixed-effect model with Šídák’s multiple comparison test was used to determine the significance of weight loss trends over the course of infection. *p*-values are indicated on plots. If no *p*-value is shown between conditions, the difference is not statistically significant.

**Figure 6 viruses-17-01225-f006:**
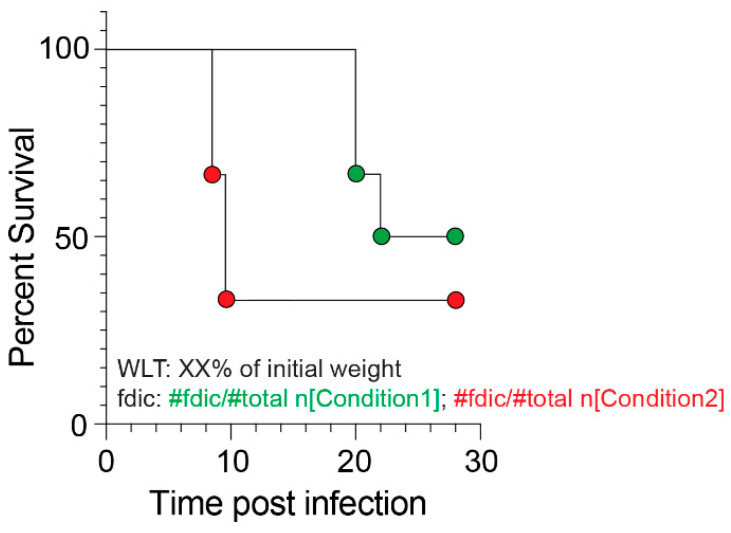
Proposed template to display mouse survival rates upon fatal viral infection. We suggest that the figure should display information about the WLT implemented, along with the number of mice fdic out of the total number of mice used per experimental condition. An illustrated example of the use of this template can be seen in Figure 2B–D. We also recommend reiterating in the figure legend the total number of mice used per experimental condition, as well as the corresponding sex ratio.

## Data Availability

The datasets generated and analyzed for this study are available from the corresponding author upon request.

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
