# Peer review of "Comprehensive Analysis of the Impact of Weight Loss Thresholds on Mouse Models of Fatal Viral Infection"

_viruses, 2025, doi:10.3390/v17091225_

Round 1

Reviewer 1 Report (New Reviewer)

Comments and Suggestions for Authors
  1. The manuscript uses mice aged 13–22 weeks (wild-type), 12–20 weeks (K18-hACE2), and 18–29 weeks (POWV). Please provide the scientific rationale for these specific age ranges.
  2. Figures 1E, 1F, and 1I include survival curves at 25% and 30% WLT, yet the corresponding weight-loss traces indicate that few or no animals reached those thresholds. Please explain how these survival curves were generated and report how many animals crossed each threshold.
  3. The Fig. 2 legend mentions “peak weight loss,” but no peak-loss panel is shown. Please add the panel or revise the legend.
  4. Overall survival in the IAV study appears high, particularly in C57BL/6J. Have the authors evaluated whether the inoculating viral titer was too low?
  5. Please clarify why C57BL/6N was not included in the IAV experiments.
  6. Please explain why a sex-bias analysis was not performed for the IAV study.

Author Response

The manuscript uses mice aged 13–22 weeks (wild-type), 12–20 weeks (K18-hACE2), and 18–29 weeks (POWV). Please provide the scientific rationale for these specific age ranges.

Our rationale was to use only adult mice for this study, that is, between 10 and 30 weeks old (neither considered young nor old).

Figures 1E, 1F, and 1I include survival curves at 25% and 30% WLT, yet the corresponding weight-loss traces indicate that few or no animals reached those thresholds. Please explain how these survival curves were generated and report how many animals crossed each threshold.

Thank you for your comment. We invite the reviewer to examine Fig. 1J, which displays the precise number of mice that reached each weight loss threshold per strain and viral inoculum. As mentioned in lines 316-321, weight loss is not necessarily coupled with lethality, which can explain some disconnect between the weight loss (mean) curve and the survival rate.  

The Fig. 2 legend mentions “peak weight loss,” but no peak-loss panel is shown. Please add the panel or revise the legend.

We appreciate this reviewer bringing this mistake to our attention. We have corrected it.

Overall survival in the IAV study appears high, particularly in C57BL/6J. Have the authors evaluated whether the inoculating viral titer was too low?

A 1000pfu dose is conventionally used in the literature to induce lethal PR8 infection. In our study, this dose proved to be significantly lethal in BALB/c mice and to cause significant disease in most animals (mean weight loss of around 15%). The lesser lethality in our C57BL/6J mice (which were directly purchased from Jackson Lab for this study) at this viral dose, as compared to some literature reports, may be due to multiple well-known variables such as mouse housing environment, infection timing etc…

Please clarify why C57BL/6N was not included in the IAV experiments.

With the need to be mindful of our resources, we aimed here to compare the impact of WLT on our two most divergent mouse strains available.

Please explain why a sex-bias analysis was not performed for the IAV study.

Thank you for this comment. We have added data related in sex in Fig1G, 4F and 5F to demonstrate that sex does not regulate the impact of the WLT on survival rate in some of our specific experimental conditions.   

Reviewer 2 Report (New Reviewer)

Comments and Suggestions for Authors

Dear Authors,

The article “Comprehensive analysis of the impact of weight loss  thresholds on mouse models of fatal viral infection” has been resubmitted after revision. The authors included in the article new viruses - IAV, Powassan Virus (POWV) and additional mouse strain (BALB/c).

But there are still serious shortcomings.

  1. It is advisable to change the Title by listing IAV, Powassan Virus (POWV), SARS-CoV-2. Such specificity is required so that the reader can better navigate when searching for data, as well as to increase citation.
  2. It is desirable to simplify the graphs, there are many unnecessary inscriptions that interfere with perception. For example, Fig. 6:

#dic/#total#n Condition 2 – what is it? There are no units of measurement on the abscissa, the graph shows the results for 30 days of the experiment, and in all experiments 15 days.

It is written in the legend of Fig6 : “We also recommend reiterating in the figure legend the total number of mice used per experimental condition, as well as the corresponding sex ratio”. What is it?

  1. In one experiment, 6-7 mice were used, in others, 9 to 16 mice, just to evaluate one parameter - weight loss thresholds. Please explain why there is such a difference in the number of mice. And how was the 3R rule observed. Were any other parameters determined that were not included in the article?
  2. The abstract and conclusion sections do not contain any remarks which could be interesting for readers, such as whether there are statistically significant differences between male and female in the recovery period. The time characteristics of weight loss are not discussed, such as why weight gain begins on day 5 with SARS-CoV-2, for example. It would be desirable to include the numerical values of the experimental results in the abstract.

Author Response

It is advisable to change the Title by listing IAV, Powassan Virus (POWV), SARS-CoV-2. Such specificity is required so that the reader can better navigate when searching for data, as well as to increase citation.

We appreciate this reviewer’s advice. However, we prefer to keep our title unchanged, as the different models we are working with in this manuscript are all mouse models of fatal viral infections. As our title also does not highlight a specific finding, but rather underscores the nature of the study that was conducted, we believe that there is no risk of our findings to be over-interpreted through our title. The fact that the other reviewers did not raise this specific point strengthens our perspective.

It is desirable to simplify the graphs, there are many unnecessary inscriptions that interfere with perception. For example, Fig. 6: #dic/#total#n Condition 2 – what is it? There are no units of measurement on the abscissa, the graph shows the results for 30 days of the experiment, and in all experiments 15 days. It is written in the legend of Fig6 : “We also recommend reiterating in the figure legend the total number of mice used per experimental condition, as well as the corresponding sex ratio”. What is it?

As described in text line 549-550 and most notably in the title and legend of Figure 6, Figure 6 is intended to serve as a template figure, that is, a suggested display of mouse survival rates.

In one experiment, 6-7 mice were used, in others, 9 to 16 mice, just to evaluate one parameter - weight loss thresholds. Please explain why there is such a difference in the number of mice. And how was the 3R rule observed. Were any other parameters determined that were not included in the article?

We appreciate this reviewer’s comment. Different numbers of mice within a specific age range were available at the time of our various experiments, explaining variations in cohort size. However, all of our mouse number provides the necessary statistical power to support our conclusions. The 3R rule was followed through this study, as it is a mandatory regulation of our institution.

The abstract and conclusion sections do not contain any remarks which could be interesting for readers, such as whether there are statistically significant differences between male and female in the recovery period. The time characteristics of weight loss are not discussed, such as why weight gain begins on day 5 with SARS-CoV-2, for example. It would be desirable to include the numerical values of the experimental results in the abstract.

We appreciate the reviewer raising these interesting questions. However, we feel they are outside the scope of our focus, that is, the impact of weight loss threshold on survival rate. We also think that our abstract should remain straightforward, and readers interested in specific numerical values can find them inside the main text sections of our manuscript.

Reviewer 3 Report (New Reviewer)

Comments and Suggestions for Authors

This is an extremely important and thorough investigation of how WLT (Weight Loss Threshold) impacts survival curves and how sex of mice, mouse strains, types of virus, and inoculum impacts the WLT ability to predict and impact survival following an infection.  This is important in evaluating treatments of viral infection, which doesn’t always depend on a decrease (or in some cases increase) in pfu/g tissue. The authors make a strong case for consideration of setting the WLT based on scientific results, rather than an arbitrary 20%, and suggest 30%.  Their data certainly support that, especially since prediction of survival by examining clinical parameters is difficult.

This is a re-review and the authors have adequately addressed the issues found in the previous review.

However, there were minor mistakes still lingering:

Figure 3A – Y axis should say “Weight Change (%)” instead of “Percent Survival”.  The figure legend says “(A) Weight loss of male.....”

Figure S2(C-D) – Line 9 on the legend should say, “Weight loss of C57BL/6J mice that survived (closed circle) or died (open circle).....”. The diagram of the circles dictates this and the current text says the opposite.

Figure S3(B) - Line 9 on the legend should say, “Weight loss of BALB/c mice that survived (closed circle) or died (open circle).....”. The diagram of the circles dictates this and the current text says the opposite.

Author Response

Figure 3A – Y axis should say “Weight Change (%)” instead of “Percent Survival”.  The figure legend says “(A) Weight loss of male.....”

We appreciate this reviewer bringing this mistake to our attention. We have corrected it.

Figure S2(C-D) – Line 9 on the legend should say, “Weight loss of C57BL/6J mice that survived (closed circle) or died (open circle).....”. The diagram of the circles dictates this and the current text says the opposite.

We appreciate this reviewer bringing this mistake to our attention. We have corrected it.

Figure S3(B) - Line 9 on the legend should say, “Weight loss of BALB/c mice that survived (closed circle) or died (open circle).....”. The diagram of the circles dictates this and the current text says the opposite.

We appreciate this reviewer bringing this mistake to our attention. We have corrected it.

This manuscript is a resubmission of an earlier submission. The following is a list of the peer review reports and author responses from that submission.

Round 1

Reviewer 1 Report

Comments and Suggestions for Authors

The manuscript by Kenney and coworkers sought to evaluate how multiple experimental variables influence the impact of weight loss euthanasia thresholds on viral virulence in preclinical studies. They focused on SARS-CoV-2 infections in mice with different experimental variables, including mouse and viral strains and the sex ratio within a mouse cohort. They successfully confirmed the correlation of body weight loss in different mouse strains with mortality after infection with different variants of SARS-CoV-2. From these results in the SARS-CoV-2 infection experiments, they defined mouse weight loss thresholds (WLT) as the percentage of weight loss triggering humane euthanasia. Overall, the availability of such a WLT is an essential factor in evaluating SARS-CoV-2 virulence in preclinical infection. However, the approach described in this manuscript to adjust the WLT to several experimental variables, including mouse strains, genetic variants of the same viral pathogen, and the sex ratio within a mouse cohort to yield resolutive data, is challenging to transfer to other studies, even in the same laboratory settings. This is due to several factors. First of all, they only use the SARS-CoV infection model. This might be difficult to apply to other viruses. The settings described in this manuscript would significantly benefit from the comparison with other viral pathogens using the same experimental variables used in the study. In addition, there are no data on SARS-CoV-2 loads in target organs, e.g., lungs. The manuscript has to include data on viral loads in organs to correlate the weight loss to the virulence of SARS-CoV-2 infection and standardize the effect of the different experimental variables. These data are essential to confirm the influence of many experimental variables –including mouse and viral strains and the sex ratio within a mouse cohort– on the ability of a WLT to yield high-resolution data.

What is the impact of different mouse strains? The manuscript used only mice with a C57BL6 background; what about Balb/c mice?

Is the outcome of the mouse-adapted SARS-CoV-2 strain MA30 wild-type mice comparable to the SARS-CoV-2 variants Delta (B.1.617.2) and Omicron (BA.1) in the k18-hACE2 mouse model?

In addition, they should rephrase their findings and conclusions to more specifically refer to their experimental setting using different SARS-CoV-2 variants. The title should also be changed. The title is too broad and general. They should include SARS-CoV-2 infection in the title. Moreover, this should be included in the whole manuscript, especially in the discussion section. Another part that should be included in the discussion is that due to different requirements from the authorities and animal welfare protection, the human end point defined for percentage of body weight loss may differ for other research institutions. That means, in some cases, the body weight can not be precisely adjusted to the specific pathogen or variables used in the study. This should be discussed in more detail.

Reviewer 2 Report

Comments and Suggestions for Authors

This manuscript hyper focuses on weight loss threshold as a major determinant for euthanasia.  It is well known that multiple variables influence morbidity and mortality in animal models, with weight loss as a singular determinant for possibility of euthanasia.  It is suggested that this manuscript be re-written to include multiple determinants that are appropriate for euthanasia criteria, as scoring systems of multiple parameters (weight loss, food/water consumption, activity levels, physical characteristics, etc.) are commonly used for euthanasia guidelines.